# A Deep Learning Method for Rolling Bearing Fault Diagnosis Based on Attention Mechanism and Graham Angle Field

**DOI:** 10.3390/s23125487

**Published:** 2023-06-10

**Authors:** Jingyu Lu, Kai Wang, Chen Chen, Weixi Ji

**Affiliations:** 1Department of Mechanical Engineering, Jiangnan University, Wuxi 214122, China; 7200832004@stu.jiangnan.edu.cn (J.L.); jndxw_kai@163.com (K.W.); 7190832003@stu.jiangnan.edu.cn (C.C.); 2Jiangsu Key Laboratory of Advanced Food Manufacturing Equipment and Technology, Jiangnan University, Wuxi 214122, China

**Keywords:** fault diagnosis, data dimensionality reduction, deep learning

## Abstract

Focusing on the low accuracy and timeliness of traditional fault diagnosis methods for rolling bearings which combine massive amounts of data, a fault diagnosis method for rolling bearings based on Gramian angular field (GAF) coding technology and an improved ResNet50 model is proposed. Using the Graham angle field technology to recode the one-dimensional vibration signal into a two-dimensional feature image, using the two-dimensional feature image as the input for the model, combined with the advantages of the ResNet algorithm in image feature extraction and classification recognition, we realized automatic feature extraction and fault diagnosis, and, finally, achieved the classification of different fault types. In order to verify the effectiveness of the method, the rolling bearing data of Casey Reserve University are selected for verification, and compared with other commonly used intelligent algorithms, the results show that the proposed method has a higher classification accuracy and better timeliness than other intelligent algorithms.

## 1. Introduction

Rolling bearings are called “industrial joints” and have important applications in providing support and reducing friction. However, it is precisely because of their application in relation to these two aspects that bearings are also one of the most vulnerable parts. Statistics show that the root cause of 45% to 55% of rotating machinery failures is associated with the failure of rolling bearings [1]. Taking common machine tool equipment as an example, rolling bearings are often used in the main shaft, ball screw, and general transmission shaft of machine tool equipment. The bearings used in the main shaft and general transmission shaft are the main transmission parts of machine tools, and their performance directly affects the speed, rotation accuracy, anti-vibration cutting performance, thermal deformation, etc., which, in turn, effects the precision and surface quality of processed parts [2].

Traditional fault diagnosis methods focus on analyzing the time-frequency domain information of the original vibration signal, in order to realize the extraction and diagnosis of fault features. The currently existing strategies are generally based on two steps of feature extraction and fault identification [3]; these are using classical signal processing methods, such as wavelet transform [4], and Fourier transform [5], to extract signal features. These classical methods often transform the original signal domain in the time domain and manually construct fault features in different domains [6]. Based on this strategy, many scholars have completed a significant amount of research. Muhammet et al. [7] used the Hilbert transform and fast Fourier transform to extract the peak frequency, specific harmonics, and amplitudes corresponding to inner ring defect (IRD) and outer ring defect (ORD) as classification features; introduced ANN (artificial neural network) and Combining GA (genetic algorithm) to establish rules for individuals in the population under the ANN system; used the backpropagation algorithm to train different randomly constructed ANN models; obtained the ANN output value through the cyclic GA algorithm; and obtained fault classification proficiently and quickly. Tang et al. [8] classified multiple time-frequency curves (MTFC) and matched the average ratio between the relevant curves with the theoretical fault characteristic coefficient to determine the fault type.

At present, intelligent fault diagnosis of bearings using traditional fault diagnosis methods has been widely recognized and applied in related fields. However, due to the limited scope of the expansion of the traditional method, a large amount of prior knowledge, and a large number of engineering examples to assist and verify, resulting in poor learning ability and robustness when dealing with complex non-linear problems and unsteady problems.

In order to avoid the possible above-mentioned problems, in recent years, in the field of fault diagnosis, the proportion of deep learning models applied to solve diagnosis problems has increased year by year. Methods such as support vector machines, artificial neural networks, and k-nearest neighbors to train and extract fault features according to specified fault conditions have been used. Wei et al. [9] used Variational Mode Decomposition (VMD) based on parameter optimization and improved Deep Belief Network (DBN) to decompose the original vibration signal in order to obtain several band-limited eigenmode components of different frequencies. By introducing the Gray Wolf optimization algorithm (GWO), the multi-scale scattering entropy is used to extract the fault feature information of the modal component with the largest correlation coefficient, and, finally, the fully connected neural network is used for fault classification and diagnosis.

The above model focuses on the direct analysis of the bearing vibration signal and its application for a one-dimensional convolutional neural network. The unique local area connection, weight sharing, and spatial sampling characteristics of the convolutional neural network demonstrate better performance when dealing with high-dimensional data.

Therefore, many scholars have turned their attention to the method of converting one-dimensional time series data into two-dimensional image data; using the characteristics of a convolutional neural network to analyze two-dimensional image data has achieved very good results. Wen et al. [10] converted the original vibration signal into a two-dimensional grayscale image, and input it into CNN for fault classification and diagnosis, which has attracted the attention of many scholars. Li et al. [11] converted the vibration signal into an image by combining the vibration acceleration signal and the integrated velocity-only displacement signal, and integrated the Dual-stage Attention-based Recurrent Neural Network (DA-RNN) and convolutional attention block module (CABM) into the training, and finally use the convolutional neural network embedded in the CBAM structure for fault classification. Zhang et al. [12] proposed a fault diagnosis method for the intelligent classification of vibration signals of different fault locations and damage degrees. By converting it into a spectrogram and retaining the original information of the time domain signal to the greatest extent, a deep fully convolutional neural network is used for fault diagnosis. Tao et al. [3] combined the short-time Fourier transform (STFT) and the categorical generative adversarial networks (CatGAN) to convert the original one-dimensional vibration signal into a two-dimensional time-frequency map, and obtained the CatGAN model generation and STFT extraction through the confrontation training process. The map organized the samples into clusters of specific categories; this method has strong robustness.

The above models all convert the bearing vibration signal into a two-dimensional image, and use the converted two-dimensional image, combined with the two-dimensional convolutional neural network, to realize the fault diagnosis of the bearing and achieve excellent detection results. However, there are still certain limitations in the diagnostic process, which makes the problem of feature information loss inevitable; at the same time, the image conversion process applied in other image conversion strategies affects the temporal correlation of the vibration signals.

Therefore, in response to the above problems, this paper proposes a method to convert the bearing vibration signal into a two-dimensional color image using the Graham angle field method. The main contributions of this study are reflected in the following aspects:This method can cover a large amount of feature information, and, in the image conversion process, the rectangular coordinate system is transformed into a polar coordinate system, and the two-dimensional image is regenerated from the converted polar coordinate system image, which can almost completely retain its temporal correlation.An improved ResNet50 model based on the attention mechanism is proposed to extract the temporal and spatial features of the image, and, finally, a classifier is used to classify and diagnose faults.On the basis of ablation experiment verification, it is proved that the method has excellent fault diagnosis performance.

The structure of this paper is organized as follows. The research related to fault diagnosis is introduced in Section 2, and the relevant theories of GAF and ResNet are explained in Section 3. Then the proposed GADF-ResNet method is explained in Section 4, and, finally, the proposed GADF-ResNet method is demonstrated in Section 5 using the classic bearing dataset. Methods were validated by ablation experiments, and the conclusions of the study are described and possible directions for future work are outlined.

## 2. Literature Review

### Data Feature Extraction in Fault Diagnosis

From the point of view of industry and academia, the early detection of equipment failures generally involves the monitoring of equipment operation data and uses feature extraction to extract information related to equipment health status from complex equipment operation data [13,14]. In particular, in recent years, with the widespread use of sensors, the sampling frequency of various signals, such as vibration, temperature, and pressure, has increased, in order to diagnose the healthy operation status of equipment and, finally, to enable the practicality of predictive maintenance [15], data analysis is necessary, and then, inevitably, the problem of data dimensionality will arise in the process. Faced with this kind of data dimensionality problem, traditional machine learning methods do not perform well. The common idea is to reduce the dimensionality of data before the data are input into the model, using methods such as artificial neural network (ANN) [16], support vector machine (SVM) [17], etc., so as to achieve better results. However, these traditional intelligent fault diagnosis methods often have poor generalization for sensitive features and excessive reliance on the prior knowledge of experts when dealing with significant amounts of data. For each specific fault diagnosis task, the feature extractor must be redesigned, therefore, it is an urgent need to research a feature selection method that can eliminate manual extraction.

Starting from the traditional autoencoder (AE), many scholars have tried various methods of deep learning in the field of fault diagnosis for fault feature recognition. For example, Jia et al. [18] used a method based on DNN (deep neural network) to find a non-linear function that can approximate complex non-linear functions from the original data, so as to find a method for automatic and accurate fault diagnosis. In addition, Xia et al. [19] used a multi-sensor fusion method combined with CNN training to avoid manual identification of features. Similarly, Liu et al. [20] used a stacked sparse AE; they combined a short-time Fourier transform and stacked sparse autoencoder to analyze the signal, through ablation. After the experiment, it has been found that this method has better performance and effectiveness. Recently, in order to better use deep learning for fault analysis of signals, Jia et al. [21] proposed a model based on amending the shortcomings of traditional autoencoders by constructing a normalized autoencoder. The locally connected network of the sparse autoencoder, by using the special rules learned from the input signal in the local layer, finally recognizes the health condition of the machine by obtaining the translation invariant features.

After the CNN was proposed, due to its advantages in extracting sensitive features from high-dimensional data, some scholars were inspired to convert vibration signals into two-dimensional images and use CNN for feature extraction, so as to finally obtain the distribution status of faults. For example, Guo et al. [22] proposed an adaptive CNN network structure which can effectively identify the features in the sample in the dataset and automatically extract deep features, the experimental verification found that it has better performance. At the same time, Zhang et al. [23] proposed a convolutional neural network with data enhancement effects that uses the width of the first convolutional layer. The kernel implements the adaptability of the model with high accuracy.

In view of the above, it can be found that deep learning has made achievements in the field of image classification, and deep learning as a feature selection method has also been widely used in equipment fault diagnosis.

For example, Shao et al. [24] designed a deep belief network (DBN) for rolling bearing fault diagnosis. They used the idea of particle swarm optimization to greatly improve the accuracy of rolling bearing fault diagnosis, and obtained a calculation result of 13 s. In addition, Chen et al. [25] used sparse encoders for fault diagnosis by fusing the time domain and frequency domain features of the vibration signal. This method has a high recognition accuracy, and the accuracy rate reaches about 98%. Methods based on the neural network have a stronger advantage in image classification, and it has become the focus of discussion by many scholars to present the sensor signal through a certain encoding method and use the neural network for feature processing. Among them, Lu et al. [26] developed a method where the signal is converted into an image using a bispectrality method to classify the image using a neural network. With 98% accuracy and robustness, the method finally achieves fault classification. Yin et al. [27] used a data generation strategy based on generative adversarial networks and convolutional neural networks to perform fault analysis on vibration signals under different noise environments, and achieved good results; however, the speed of evaluation decreases significantly with the increase in sample size. 

Although CNN has demonstrated significant advantages in terms of image feature extraction, there is still room for further improvement in relation to the problem of feature extraction from two-dimensional images converted from bearing vibration signals. This paper aims to provide possible improvements and solutions to this problem.

First, the Graham angle field diagram, obtained by passing the original vibration signal through the Graham angle field, is used as the input vector feature of the neural network. However, this kind of simple signal conversion as a one-dimensional input of fault diagnosis still cannot obtain high-dimensional information. In order to better understand and mine the high-dimensional information of faults and improve the accuracy and speed of neural network recognition, this paper introduces the attention mechanism in ResNet. In the process of feature extraction, we refer to the SeBlock module. According to the influence of feature channels on model performance, different weight factor values are assigned to different feature channels.

Through this method, the discriminative feature channels in the image are highlighted, and the inconspicuous features in the image are suppressed, thereby enhancing the pertinence of the model to specific channels, and finally improving the fine feature extraction and local area localization capabilities of the model.

## 3. Materials and Methods

### 3.1. GAF Timing Recoding

Gramian angular field (GAF) [28,29] is a data conversion method that can convert one-dimensional time series data into two-dimensional image data, which ensures that the time dependence and time correlation of the original time series signal both have a good degree of restoration. In this paper, the Graham angle field method is used to convert the vibration signal data of the bearing into image data, and the realization process is as follows.

***Step 1:*** Normalization

First, integrate all the values of the collected vibration signal data $X=\{x_{1},x_{2},\cdots ,x_{n}\}$ into the interval [−1, 1] or [0, 1] through Formula (1).
(1)$$\left\{\begin{array}{l} {\dot{x}}_{-1}^{i}=\frac{\left(x_{i}-\max\left(X\right)\right)+\left(x_{i}-\min\left(X\right)\right)}{\max\left(X\right)-\min\left(X\right)},1\leq i\leq n\\ {\dot{x}}_{0}^{i}=\frac{x_{i}-\min\left(X\right)}{\max\left(X\right)-\min\left(X\right)},1\leq i\leq n \end{array}\right.$$

***Step 2:*** Timing Recoding

For the time series after integration, re-encode and locate in the polar coordinate system according to Formulas (2) and (3), and map the time series to polar coordinates in combination with Formulas (2) and (3).
(2)$$\varphi_{i}=\arccos\left({\dot{x}}_{i}\right),-1\leq {\dot{x}}_{i}\leq 1,{\dot{x}}_{i}\in \dot{X}$$
(3)$$r_{i}=\frac{t_{i}}{N},t_{i}\in N$$
where, Formula (2) recodes the normalized data into angle information, φ is the encoded angle. Equation (3) is based on the positioning of the timestamp and recodes it as the distance from the positioning point to the origin of the polar coordinates, where $t_{i}$ is the corresponding timestamp of $x_{i}$ is the total length of the time series. Combining the angle in Formula (2) and the distance in Formula (3), a unique mapping result can be generated in the polar coordinate system. At the same time, the image generated according to Formulas (2) and (3) can guarantee better time correlation, and it is also superior to other similar methods in terms of richness of feature information.

***Step 3:*** Image Conversion

On the basis of the generated polar coordinate system diagram, we use the angle sum or difference between different positioning points to generate Gramian Angular Summation Field (GASF) and Gramian Angular Difference Field (GADF). The coding formula is a combination of Formula (4) and (5).
(4)$$\begin{array}{l} A_{GSF}\left(i,j\right)=\left[\cos\left(\varphi_{i}+\varphi_{j}\right)\right]=\\ \left[\begin{array}{ccc} \cos\left(\varphi_{1}+\varphi_{1}\right) & \cdots & \cos\left(\varphi_{1}+\varphi_{n}\right)\\ \cos\left(\varphi_{2}+\varphi_{1}\right) & \cdots & \cos\left(\varphi_{2}+\varphi_{n}\right)\\ \vdots & & \vdots \\ \cos\left(\varphi_{n}+\varphi_{1}\right) & \cdots & \cos\left(\varphi_{n}+\varphi_{n}\right) \end{array}\right] \end{array}$$
(5)$$\begin{array}{l} A_{GDF}\left(i,j\right)=\left[\sin\left(\varphi_{i}-\varphi_{j}\right)\right]=\\ \left[\begin{array}{ccc} \sin\left(\varphi_{1}-\varphi_{1}\right) & \cdots & \sin\left(\varphi_{1}-\varphi_{n}\right)\\ \sin\left(\varphi_{2}-\varphi_{1}\right) & \cdots & \sin\left(\varphi_{2}-\varphi_{n}\right)\\ \vdots & & \vdots \\ \sin\left(\varphi_{n}-\varphi_{1}\right) & \cdots & \sin\left(\varphi_{n}-\varphi_{n}\right) \end{array}\right] \end{array}$$

The above formulations show different matrix representations for the transformation of a 1D signal into a 2D image. Below, we use part of the sensor signal for image conversion, and the conversion process is shown in Figure 1. As shown in Figure 1, GADF is richer in color than GASF images and contains more detailed information. This article will use GADF image coding methods for image conversion.

### 3.2. Improved ResNet50 Model Based on Attention Mechanism

In recent years, with the rapid development of deep learning technology, the network structure has gradually deepened, and the expression ability and learning ability of the model have also been enhanced. However, it has inevitably led to the emergence of problems, such as gradient disappearance or gradient explosion and model degradation. In order to optimize such problems, the following methods are often involved, such as initialization weights, batch normalization, and stochastic gradient descent. These methods can achieve certain effects, but when the number of network structure layers reaches a certain limit, it will still fail. The ResNet model proposed by He et al. [30] in 2015 perfectly solved this problem by introducing the residual block structure, and realized that the number of network layers can reach more than 1000 layers under the premise of ensuring the model has high accuracy.

The residual block structure can be divided into two types, Basic Block and Bottleneck Block, according to different application environments, as shown in Figure 2 By introducing a shortcut branch, the low-level information is shared to the high-level through layer-skip connections, which solves the problem that the gradient cannot be optimized during the backpropagation of the deep network, and breaks through the model degradation and gradient anomalies that limit the number of model layers. 

For different residual block structures, the Bottleneck Block has fewer parameters than the Basic Block by changing the number of channels, and is more economical in terms of calculation, and is often used in networks with deeper layers. 

According to the difference in residual block structure, ResNet can be divided into the following common models, ResNet18, ResNet 34, ResNet50, ResNet101, and ResNet152. Considering the device configuration and the difficulty of image training, this article uses ResNet50 as the basic model, and the parameters are shown in Table 1 and Figure 3

The ResNet structure diagram can refer to the following figure. A 7 × 7 convolution kernel is responsible for feature extraction, and the convolution step is 2 so that the length and width of the image are reduced to 1/2 of the previous image. Then, we apply a MaxPooling layer to further reduce the resolution of the image.

By using the residual block, the number of input feature map channels is doubled. 

Each stage will consist of a down sampling block and two residual blocks. The down sampling is set to the initial convolution step size, which is 2; by this method, the length and width can be reduced. In the residual block, by setting the convolution parameters, the characteristics of the input and output of the residual block can be controlled to be consistent, so that they can be added, which avoids solving the gradient disappearance and gradient descent problems of deep networks. Most of the structure of ResNet is 1 layer Conv + 4 blocks + 1 layer fc, which is divided into 5 stages, as shown in Table 1, line 2 to line 8.

At the very beginning, ResNet preprocesses the input, that is, the first line in the table (256 × 256 × 3) represents the input width, height, and number of channels. At this stage, it includes the following three successive operations.

Through the convolution kernel with a size of 7 × 7 and 64, after batch normalization it is activated using the ReLu activation function.

The second layer at this stage is the maximum pooling layer. The maximum pooling layer performs convolution operations through a kernel with a size of 3 × 3 and a step size of 2. The output image size is (64 × 64 × 64), where 64 is the number of convolution kernels in the first convolutional layer. In general, at this stage, the input with a shape of (256 × 256 × 3) passes through the convolutional layer, BN layer, ReLU activation function, and MaxPooling layer to obtain an output of (64 × 64 × 64).

After going through the above description, it is easier to understand the subsequent stages in the table. The bottleneck consists of 4 variable parameters, the number of channels and the length and width. The input shape X is defined as (C, W, W) through BN and ReLU as the convolution function F(x), and the two are added to F(x) + x, the output is obtained through a ReLU function, and the shape of the output is still (C, W, W). Through this operation, the dimension difference between the input and the output is matched, and then summed.

### 3.3. SeBlock Module Based on Attention Mechanism

In order to solve the problem of redundant information in the signal, this paper uses the SeBlock module [31] in the feature extraction stage. This module adopts the feature recalibration strategy, that is, according to the degree of influence of the feature channel on the performance of the model, different feature channels are given weighting factors of different values and take them into account in the model. By introducing this module, the optimal selection of feature channels and the maximum utilization of model performance can be achieved. Its principle is shown in Figure 4.

As shown in Figure 4, we input the image to be processed, the size of the image is $h\times w\times c_{1}$, h and w are the length and width of the image, and $c_{1}$ are the number of channels of the image. After a $F_{tr}$ convolution operation, a feature map with $c_{2}$ channels are generated. Then, global average pooling is performed on the h×w dimension of the feature map. After pooling, the size of the feature map becomes $1\times 1\times c_{2}$, and the resulting feature map is fully connected to obtain a fully connected layer of $1\times 1\times c_{2}$ size. After the Sigmoid operation, the weighted operation is performed with the corresponding channel of the feature map. After the above processing, the model realizes the self-adaptive secondary calibration channel influence factor, which highlights the discriminative feature channels in the image, suppresses the non-significant feature channels in the image, and enhances the pertinence of the model for specific channels.

### 3.4. Classification Network Model Based on Attention Mechanism

The input data of the model in this paper is the GADF image obtained by recoding the vibration time series signal. In order to enable the model to have better performance, this paper introduces the SeBlock module to increase the fine feature extraction and local area targeting capabilities of the model. The model uses the Bottleneck Block in ResNet as the basic framework, and incorporates channel domain attention on this basis. The model framework structure is shown in Figure 4. According to the size of the feature image being processed, the framework can be divided into hierarchical modules of conv2_x, conv3_x, conv4_x, and conv5_x in the figure. Due to the use of multi-scale feature maps, the insufficient semantic information of the channel that has a greater impact on the classification results will lead to poor classification results. In order to solve this problem, this paper introduces an attention module to realize the adaptive adjustment of feature weights. As shown in Figure 5, the feature images of each scale are deconvoluted and convolved to the same size, and then the SeBlock module is introduced to adjust the corresponding weights. Then, the image is input to the Avg Pool layer for global average pooling, and is, finally, classified in the FC layer.

### 3.5. Fault Diagnosis Method of Rolling Bearing Based on GAF-SEResNet

The problem of bearing fault diagnosis is an extracted feature classification problem, and the deep residual network model is currently the best classification network model used to solve the problem. However, the ResNet network model lacks good local targeting capabilities. It is easy to fall into the dilemma of unbalanced weight distribution. In view of the shortcomings of the traditional ResNet model, this paper proposes a GADF-SEResNet model to solve the above problems. On the basis of ResNet50, the model adjusts the weight adaptively by combining the attention mechanism and mutual fusion strategy, so as to improve the accuracy of bearing fault diagnosis. The specific method flow chart is shown in Figure 6.

***Step 1:*** Collect the original vibration signal of the bearing and generate corresponding data files for model training.***Step 2:*** The collected signal is divided into data according to the sample length by overlapping sampling, and corresponding data samples are generated.***Step 3:*** Reconstruct the generated data sample into a two-dimensional feature image according to the GADF encoding method, and construct an image dataset accordingly.***Step 4:*** Divide the feature image dataset into a training set and a test set according to a certain ratio.***Step 5:*** Build an improved SEResNet neural network model and initialize network parameters.***Step 6:*** Input the training set into the model for pre-training, and optimize the model parameters until the optimal value is reached, then proceed to Step 7, otherwise re-execute Step 5, adjust the model parameters until the optimization effect reaches the best, and save the current model.***Step 7:*** Input the test set data into the trained improved SEResNet model for rolling bearing fault diagnosis, and then obtain the bearing fault classification result and classification accuracy.

## 4. Experimental Research and Analysis

### 4.1. Experimental Dataset

In order to verify the correctness of the method, this paper will use the bearing vibration experimental data released by the Bearing Vibration Data Center of Case Western Reserve University (CWRU) [32]. Experiments were carried out with the data of the drive end bearing. The bearing model was SKF 6205-2RS deep groove ball bearing. The single point damage was arranged by EDM on the inner ring, outer ring, and rolling elements of the bearing. The damage diameter was 0.1778 mm and 0.3556 mm, and 0.5334 mm, across a total of nine fault states. An acceleration sensor is placed above the bearing seat to collect vibration signals, and the sampling frequency is set to 12 kHz. In this paper, the vibration signals and normal vibration signals of different damage positions when the damage diameter is 0.1778 mm are selected as the research objects. Corresponding data samples were made for these bearings in different states.

#### 4.1.1. Data Processing

This article uses SEResNet as the main optimization model, which requires a large number of data samples. Based on the demand for a large amount of data, this article performs data enhancement operations in the data processing stage to ensure the accuracy of model optimization. The experimental data collection environment is that the bearing speed is 1772 r/min, and the signal sampling frequency is 12 kHz. According to the formula $N=f_{s}\times 60/n$, it can be calculated that the number of sampling points corresponding to one cycle is 406. According to the strategy that each sample contains at least one complete vibration cycle and maximizes the number of samples, the sample length is determined to be 1024 data points, and the step size is 512 data points. Data enhancement is performed by overlapping sampling. Half of the signal area overlaps with the latter sampling signal, so that the signal can be fully utilized and the number of samples can be further enriched. After the data are segmented in an orderly manner according to the corresponding sampling method, the segmented one-dimensional vibration signal sample is labeled. Each type of signal has 1000 samples, including a total of 4000 samples, which are divided into a training set and a test set according to the ratio of 8:2. The specific fault sample distribution is shown in Table 2.

#### 4.1.2. Feature Image Generation

Convert the sample that has been segmented and sampled into a two-dimensional feature image through the Gram angle field, and the image resolution is set to 256×256. In order to characterize different types of faults, the typical feature images of four different faults are shown in Figure 7. It can be seen that the texture of the feature images transformed by the Graham angle field is clear and the features are obvious, which is beneficial for fault classification.

### 4.2. Model Training and Analysis of Training Results

The deep learning framework used in the experiment is Python-based Pytorch1.12.0, CUDA 11.6, CUDNN 8.4.0, the computer processor model used is Intel(R) Core (TM) i7-10875H CPU v16, and the graphics card model is the GPU NVIDIA GeForce RTX 2060, the memory size is 16 GB, the Adam optimizer is used to adaptively optimize the parameters, and the small-batch stochastic gradient descent method is used to train the model. Considering the computer processing power, the batch size is selected to be 8, the initial learning rate is 0.001, and the training iteration round is set to 80.

We input the divided dataset into the model for training, and after 80 rounds of training iterations, the accuracy and Loss function are obtained, as shown in Figure 7. Analysis of the results in Figure 8 shows that after 40 iterations of the model, the accuracy and loss values tend to be stable until the training is completed and the model reaches full convergence. The accuracy rate of the training set reached 98.63%, the accuracy rate of the test set reached 98.37%, and the training loss decreased to 0.015 and stabilized. It can be seen that the GADF-improved SEResnet model proposed in this paper has good classification and diagnosis results.

In order to verify the classification recognition ability of the model, the confusion matrix is drawn for quantitative analysis according to the classification results. The confusion matrix is shown in Figure 9. The horizontal and vertical coordinates in the figure correspond to the actual category and the predicted category, respectively, and the values on the main diagonal of the matrix correspond to the proportion of correct classification of each type of fault. The darker the color of the value block, the higher the prediction accuracy. It can be seen from Figure 9 that three samples of the first category and the second category are misclassified to other categories, only one sample of the third category is misclassified to the fourth category, and five samples of the fourth category are predicted to appear. The overall average classification accuracy rate is 98.37%, which reflects that the GADF-SEResNet model has played a significant role in solving the problem of bearing fault diagnosis.

### 4.3. Comparative Analysis with Other Image Coding Methods

Throughout the above experiments, it can be seen that the improved ResNet model, with the introduction of an attention mechanism, can well realize the fault diagnosis of bearings. In order to further verify the reliability of the image coding method, other image coding methods are combined with the improved model in this paper for comparative verification. In [33], the vibration signal is subjected to continuous wavelet transform (CWT) to construct a time-frequency map (TFR), which is compressed to an appropriate size and then input to CNN as a feature map for rolling bearing fault diagnosis. Literature [34] converts the one-dimensional vibration signal into a grayscale image (GI), and on this basis, combines the auxiliary classification to generate an adversarial network, uses the label of the original data as the input of this network for data enhancement, and inputs it to CNN for bearing fault classification. Four different timing signal fault samples are recoded into corresponding feature images by the method in [15,16], and compressed to the size of 256×256. We input the processed feature image to the neural network model in this paper for training, and the training results are shown in Table 3.

It can be seen from Table 3 that the accuracy of the model combined with the time-frequency image and the grayscale image is 96.36% and 95.42%, respectively, which is slightly lower than the accuracy of the combined Graham angle field image. From the perspective of the number of parameters, since the feature image of the Graham angle field contains more feature information than the time-frequency image and the gray-scale image, the number of parameters is more than the time-frequency image and the gray-scale image. At the same time, the training time and classification accuracy of the Graham angle field feature image are better than the time-frequency image and grayscale image. It can be seen that the image coding method using the Graham angle field can better extract feature information, and it is more suitable for bearing fault classification and diagnosis compared with other image coding methods.

### 4.4. Comparative Analysis with Other Algorithm Models

In order to verify the effectiveness of the algorithm model used in this paper in solving bearing fault classification and diagnosis problems, the algorithm model in this paper is compared with other algorithm models, as shown in Table 4.

After changing the classification model to 1DCNN, the accuracy rate has been improved to a certain extent, but it remains in the range of 70% to 80%, which cannot meet the requirements for actual use. Corresponding to the above method, after recoding the one-dimensional vibration signal into a GADF image and inputting it into the deep learning model, the accuracy rate has been significantly improved, and the accuracy rate of each model has remained above 85%. It can be seen from this that converting the vibration signal into a GADF image has a significant effect on improving the overall recognition accuracy of the model. The model in this paper has the highest accuracy rate among all models using the GADF image coding method, which is about 5% higher than the original unimproved ResNet model. At the same time, the number of model parameters in this paper is less than that of the original unimproved ResNet model, and the training time is also reduced. This paper also introduces GhostNet [35] and the VGG16 model [36] for comparative experiments. GhostNet is a lightweight neural network docked on the mobile terminal. In contrast, the GhostNet model has far fewer parameters than other similar deep learning models, and the training time is also greatly shortened, but its accuracy rate is 91.54%, which is slightly lower than the model in this paper.

The VGG16 model has too many convolutional layer channels, and its number of parameters is far more than other models. It requires significant amounts of computer storage and time during training. As shown in the table, compared with other similar deep learning models, the VGG16 model training consumes the most time, with an accuracy rate of 85.72%.

In summary, the GADF-improved SEResNet model can well extract and retain the characteristic information of the input data, and realize the accurate classification and diagnosis of bearing faults to the greatest extent.

## 5. Conclusions

This paper proposes a new data mining method for bearing fault diagnosis. The main contributions of this new method include the following.

First, the bearing vibration signal is converted into a two-dimensional image by using the Gram angle field. This method can restore the correlation between the signal and time in the time series signal, and on this basis reveal the health status of the bearing vibration signal.

This visual transcoding method performs time series relationship mapping in the polar coordinate system, that is, it re-encodes the signals one by one according to the positioning point of the time stamp in the polar coordinate system in order to generate unique mapping. This structure reconstructs the one-dimensional vibration signal into two-dimensional image space so as to further mine the characteristic attributes from them.

Different from the traditional convolutional neural network, the proposed SEResNet combines the characteristics of each channel of the input image with different weight factors, and combines the characteristics of ResNet50 to maximize the selection of feature channels, and on this basis introduces the SeBlock module to extract features from local areas of the network to improve. Using the above method, the mined data features are more robust and have more precise detail features.

The effectiveness of GAF-SEResNet is verified by the CWRU bearing vibration experimental data, and the proposed GAF-SEResNet model performs well in solving the problem of rolling bearing fault diagnosis. The accuracy rate of its algorithm is 98.37%, which is higher than other image coding methods and other algorithm models. It has better feature extraction ability and good generalization performance, which shows that the proposed feature mining method can be suitable for the regression analysis model of the bearing fault type mapping relationship, and can meet the needs of machining monitoring.

## Figures and Tables

**Figure 1 sensors-23-05487-f001:**
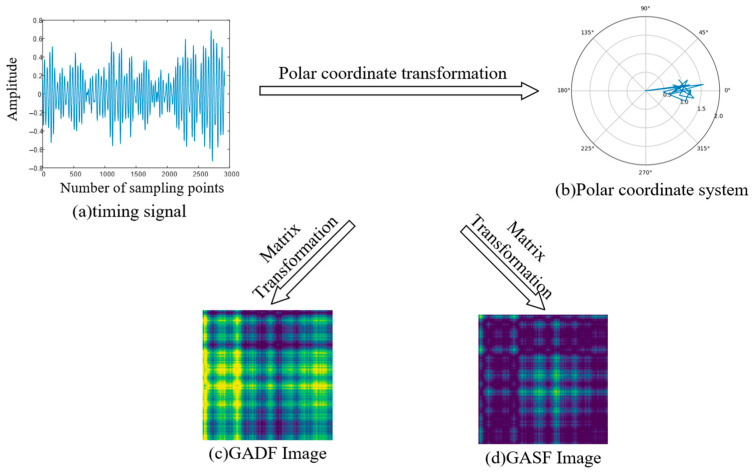
GAF encoding process.

**Figure 2 sensors-23-05487-f002:**
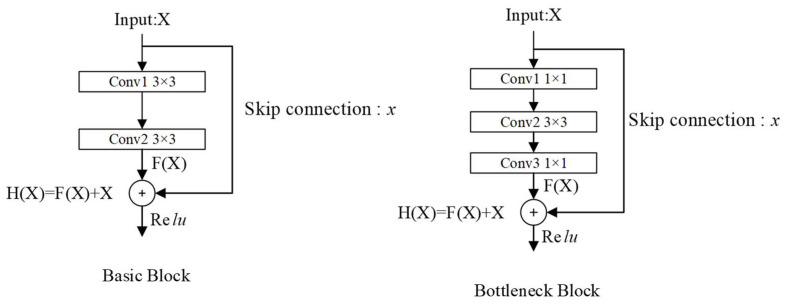
Residual network structure.

**Figure 3 sensors-23-05487-f003:**
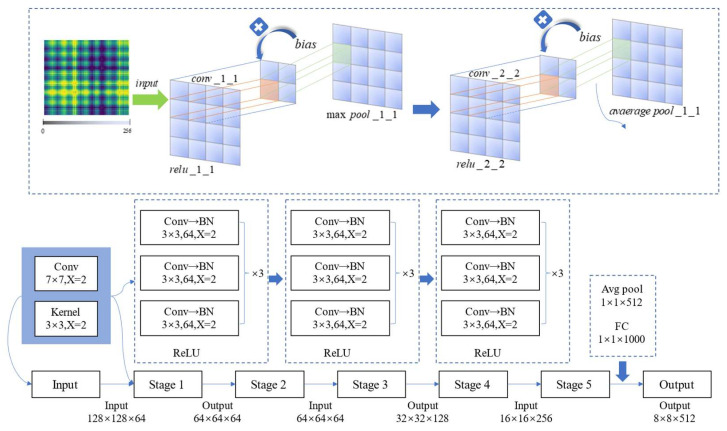
Overall residual network structure.

**Figure 4 sensors-23-05487-f004:**
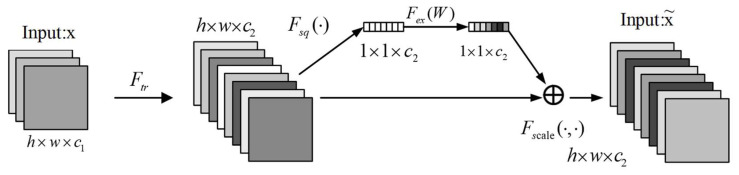
Channel domain attention mechanism.

**Figure 5 sensors-23-05487-f005:**
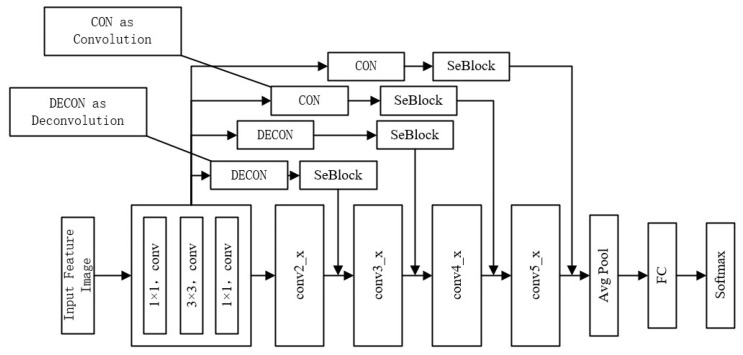
Model frame diagram used in this paper.

**Figure 6 sensors-23-05487-f006:**
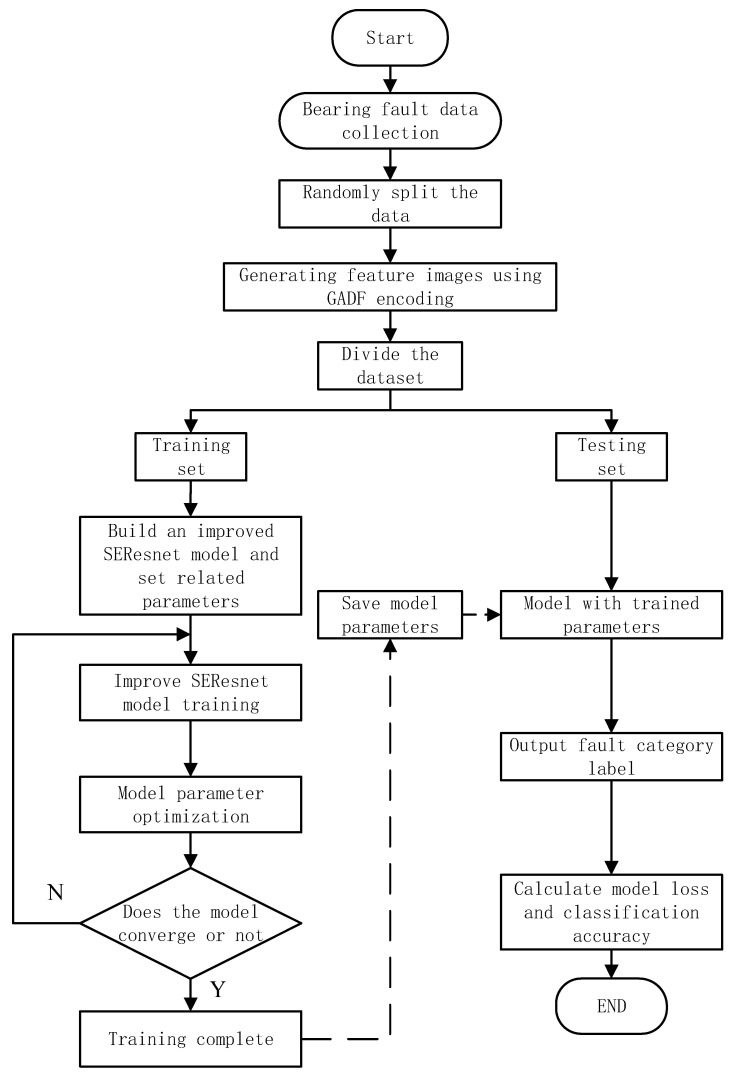
Diagnosis flow chart of rolling bearing fault based on GADF-improved SEResNet.

**Figure 7 sensors-23-05487-f007:**
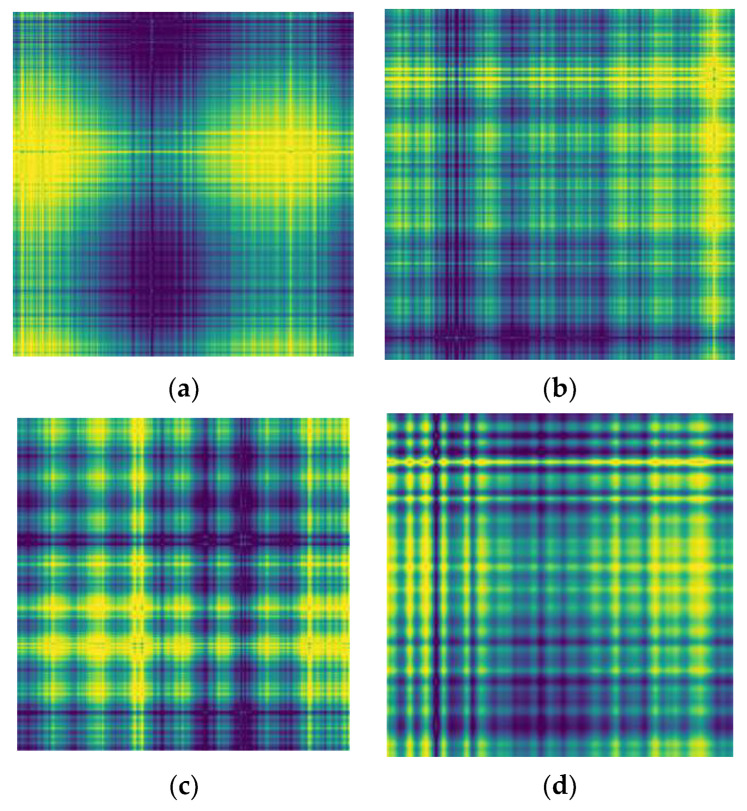
GADF coding diagram of various fault states: (**a**) normal; (**b**) slight damage to the inner ring; (**c**) slight damage to the outer ring; and (**d**) slight damage to rolling elements.

**Figure 8 sensors-23-05487-f008:**
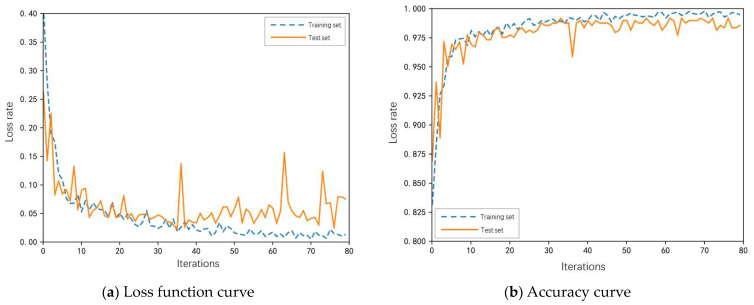
Accuracy and loss function curve: (**a**) loss function curve; (**b**) accuracy curve.

**Figure 9 sensors-23-05487-f009:**
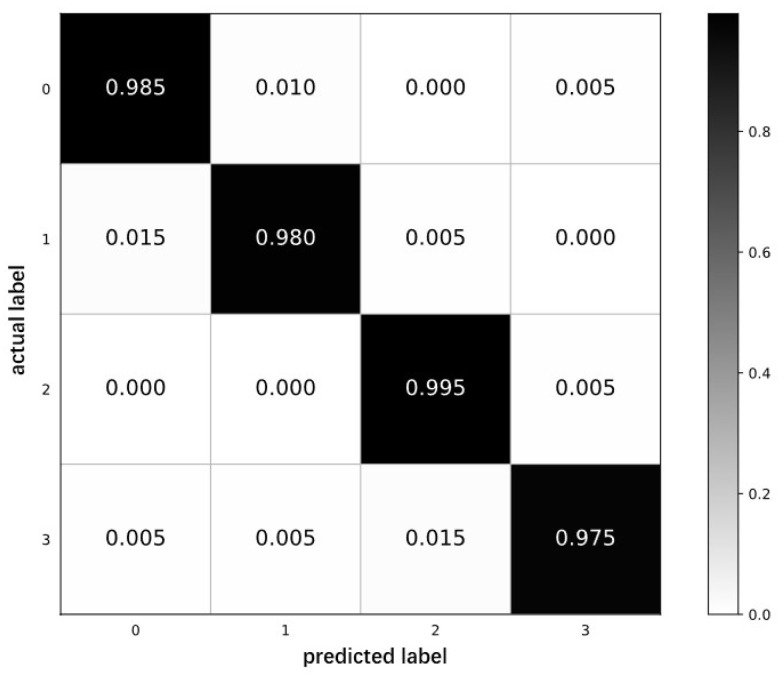
Confusion matrix.

**Table 1 sensors-23-05487-t001:** ResNet50 model parameters.

Serial	Layout	Output Dimension	Hierarchical Design
1	Input	256×256×3	-
2	Conv1	128×128×64	7×7×64,x=2
3	BN, ReLU	128×128×64	-
4	Max Pool	64×64×64	3×3,x=2
5	Conv2_x	64×64×64	$\left[\begin{array}{cc} 1\times 1 & 64\\ 3\times 3 & 64\\ 1\times 1 & 256 \end{array}\right]\times 3$
6	Conv3_x	32×32×128	$\left[\begin{array}{cc} 1\times 1 & 64\\ 3\times 3 & 64\\ 1\times 1 & 256 \end{array}\right]\times 4$
7	Conv4_x	16×16×256	$\left[\begin{array}{cc} 1\times 1 & 64\\ 3\times 3 & 64\\ 1\times 1 & 256 \end{array}\right]\times 6$
8	Conv5_x	8×8×512	$\left[\begin{array}{cc} 1\times 1 & 64\\ 3\times 3 & 64\\ 1\times 1 & 256 \end{array}\right]\times 3$
9	ReLU	8×8×512	-
10	Avg pool	1×1×512	8×8,x=1
11	Fc	1×1×1000	-
12	softmax	1000	-

**Table 2 sensors-23-05487-t002:** Data sample.

Fault Location	Damage Diameter/mm	Training Sample	Test Sample	Label
Normal	0	800	200	0
Inner Circle	0.1778	800	200	1
Outer Ring	0.1778	800	200	2
Rolling Element	0.1778	800	200	3

**Table 3 sensors-23-05487-t003:** Fault diagnosis test results of different image coding methods.

Layout	Number of Parameters/Pieces	Training Time/ms	Accuracy/%
TFR	25,046,021	14,285	96.36
GI	24,629,431	14,325	95.42
GADF	25,328,635	13,862	98.37

**Table 4 sensors-23-05487-t004:** Fault diagnosis test results of different network algorithms.

Method	Number of Parameters/Pieces	Training Time/ms	Accuracy/%
1DCNN + Signal	15,467,261	6458	78.56
GADF + VGG16	136,145,681	34,436	85.72
GADF + GhostNet	4,724,368	3524	91.54
GADF + ResNet50	26,541,072	15,241	93.22
GADF-SEResNet	25,328,635	13,862	98.37

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
