# Peer review of "A Deep Learning Method for Rolling Bearing Fault Diagnosis Based on Attention Mechanism and Graham Angle Field"

_sensors, 2023, doi:10.3390/s23125487_

Round 1

Reviewer 1 Report

- 80% training set and 20% test set? Why is the training set so big?

- What is the structure of the ANN network (number of layers, number of neurons, transfer function)?

- Why is the number of iterations limited to 80?

- It is known that the choice of optimizer can significantly impact the speed and quality of convergence during training, as well as the final  

  performance of the deep learning model. Have the authors tried other deep-learning optimizers besides ADAM optimizer?

- How suitable is the GADF-SEResnet model for early diagnosis of problems at fault locations? What is it

  about diagnostics when we have multiple faults at the same time? How does the model classify Labels in that case?

Author Response

Response to Reviewer 1 Comments

Point 1: 80% training set and 20% test set? Why is the training set so big?

Response 1: Thank you for your comment. For the traditional machine learning stage (the data set is on the order of 10,000), the general distribution ratio is 7:3 or 8:2 for the training set and test set.

However, this division method is indeed arbitrary. In order to further test the suitability of the divided data set, the obtained recall rate and accuracy are shown in the following table:

Training

Precision

Recall

0.8

0.3895

0.1652

0.6

0.3744

0.1370

0.4

0.3686

0.1005

0.2

0.2563

0.0834

Point 2: What is the structure of the ANN network (number of layers, number of neurons, transfer function)?

Response 2: Thank you for your comment. Please check out the structure of the network in the Word file.

First of all, due to the deepening of the network structure in deep learning, although the expression ability and learning ability of the model are also constantly improving, as the network depth continues to deepen, the problem of gradient disappearance or gradient explosion will inevitably occur.

That is to say, at present, the methods for optimizing the neural network are all based on the idea of backpropagation, that is, the error calculated according to the loss function is backpropagated through the gradient to guide the update and optimization of the deep network weights.

Stacked by many nonlinear layers, each layer of nonlinear layer can be regarded as a nonlinear activation function, so the entire deep network can be regarded as a composite nonlinear multivariate function.

From the point of view of the activation function, based on the improved tanh function of the traditional activation function, its derivative diagram is as follows:

As the number of network layers continues to superimpose, the final calculated gradient update will increase exponentially, that is, a gradient explosion will occur. If this part is less than 1, then as the number of layers increases, the calculated gradient update information will increase exponentially. Formal decay, that is, gradient vanishing occurs.

The ResNet model perfectly solves this problem by introducing the residual block structure, and realizes that the number of network layers can reach more than 1000 layers under the premise of ensuring the model has high accuracy.

By introducing the shortcut branch, the low-level information is shared to the high-level through layer-skip connections, which solves the problem that the gradient cannot be optimized during backpropagation of the deep network, and breaks through the model degradation and abnormal gradient problems that limit the number of model layers. For different residual block structures, Bottleneck Block has fewer parameters than Basic Block by changing the number of channels, and is more economical in terms of calculation, and is often used in networks with deeper layers. According to the difference in residual block structure, ResNet can be divided into the following common models: ResNet18, ResNet 34, ResNet50, ResNet101, ResNet152. Considering the equipment configuration and the difficulty of image training, this paper uses ResNet50 as the basic model And the parameters is described as Table. Please check out the Table in the Word file.

Point 3: Why is the number of iterations limited to 80?

Response 3: Thank you for your question. We found through experiments that if there are too many training times, there will be underfitting phenomenon.

From the above figure, we found that the number of fitting starts at 80 times, and the training effect gradually stabilizes. By observing the fitting effect of 80-120 times (marked by the part in the figure), in order to ensure the timeliness of training and significantly improve the training effect speed, we chose to use 80 instead of 120.

Point 2: It is known that the choice of optimizer can significantly impact the speed and quality of convergence during training, as well as the final  performance of the deep learning model. Have the authors tried other deep-learning optimizers besides ADAM optimizer?

Response 4: Thanks to the your suggestions. In fact, in the process of conducting experiments, we have tried to use other optimizers, including some traditional SGD and SGD variants such as SGD with Momentum, but it seems that these optimizers cannot be well integrated with GADF graphs. Although SGD may achieve higher accuracy in various optimization scenarios than Adam optimizer, its timeliness is relatively poor, and various bugs often occur when using it.

Point 5: How suitable is the GADF-SEResnet model for early diagnosis of problems at fault locations? What is it about . when we have multiple faults at the same time? How does the model classify Labels in that case?

Response 5: Thank you for your question. At present, we can only analyze the vibration signal data set over a period of time through the bearing vibration signal collected by the vibration test, so as to obtain the final diagnosis result. Your suggestion is very helpful. We will conduct further research on the early signal online diagnosis in the future work, and transmit the signal through Json through Web, but how to ensure the synchronization of signal timing during the transmission process is also our focus to break through the problem, we plan to design a microservice-based architecture to ensure the real-time performance of the signal. Please check the specific research in our upcoming paper.

Reviewer 2 Report

A fault diagnosis method for rolling bearings based on Gramian angular field (GAF) coding technology and improved ResNet50 model was proposed. Using the Graham angle field technology to recode the one-dimensional vibration signal into a two-dimensional feature image, using the two-dimensional feature image as the input of the model, combined with the advantages of the ResNet algorithm in image feature extraction and classification recognition, to realize automatic feature extraction and fault diagnosis, and finally achieve the classification of different fault types. In order to verify the effectiveness of the method, the rolling bearing data of Casey Reserve University is selected for verification, and compared with other commonly used intelligent algorithms, the results show that the proposed method has higher classification accuracy and better timeliness than other intelligent algorithms. The following issues should be answered.

Why is GAF better than traditional wavelet transform in the classification of two-dimensional time-frequency images? Can you explain the underlying mechanism?

 In addition to a brief introduction to the characteristics of the model, the basis for parameter selection of various machine learning models compared in this paper should be given, as well as the realization and residual error change of the model in the training set and the test set.

 Table 4 is the same as Table 3. Classification results of different classification models are not given.

Author Response

回复审稿人 2 条评论

要点一:为什么GAF在二维时频图像的分类上优于传统的小波变换?你能解释一下潜在的机制吗?

响应 1:谢谢你的问题。正如您所说,小波变换作为一种经典的信号处理方法已经被广泛使用。刚开始写这篇论文的时候,我们并没有考虑经典的信号处理方法,而是想找到一种数据降维的方法来处理信号,而在工厂场景中,面对的信号可能是多样的、多源异构的。我们要达到一个目标,我们可以通过结构化和非结构化数据来识别多源异构数据,对于具有挖掘内部信息能力的深度学习,所以我们选择使用基于深度学习的方法来处理信号,首先我们希望将信号以方便的方式转换为图像,并将处理后的轴承振动信号图像集成到多模态数据中。在这,

小波变换在处理振动信号方面具有其他方法无法比拟的高度。它可以通过相空间重建等技术将一维信号转换为二维图像。我们认为这种方法还是不够快,因为振动信号在设备运维过充的故障诊断中只能作为一部分数据源不能全部输入,尤其是面对生产线过程,在除了考虑故障,还需要考虑车间场地、产品方案等多方面因素,才能最大限度地提高设备运维效率,从而最终保证生产效率。

另外,虽然一维信号可以通过RNN等方法处理时间序列,CNN在处理图像方面更有优势,但是CNN作为传统的图像特征提取方法还是有不足的,所以为了能够更快速的提取故障信号的特征,改进ResNet网络注意力机制的参数。通过在初始卷积层和每个卷积层的连接处加入attention机制,对处理后的二维图像进行特征提取,请查看Word文件中的图。

随着attention机制的接入,我们发现改进后的ResNet在使用过程中的速度和准确率都有了很大的提升。但是,要完成我们最初的蓝图,我们的工作还有很长的路要走。

我们一直在努力寻找一种方法来融合信号和图像,并在深度学习中进一步处理它们。在我们未来的研究中,我们将进一步改进这种多模态特征提取方法,以实现数据预处理。信号与图像融合步骤完成,最终可以实现多模态数据的特征提取,供设备运维使用。

Point 2:  除了简单介绍模型的特点外,还应给出本文对比的各种机器学习模型的参数选择依据,以及模型在训练集中的实现和残差变化和测试集。

回复2:感谢您的评论。我们在手稿(第8页)中仔细添加了参数说明,并添加了更多关于我们在本手稿中选择的ResNet50技术路线的描述,以使其易于阅读。请查看我们的最新版本或阅读以下内容:

ResNet结构图可以参考下图。一个7*7的卷积核负责特征提取,卷积步长为2使得图像的长宽先缩小到之前的1/2。然后通过一个 MaxPooling 层进一步降低图像的分辨率。

通过使用残差块,输入特征图通道的数量增加了一倍。

每个阶段将由一个下采样块和两个残差块组成。下采样设置为初始卷积步长,即2. ,这样可以减少长宽。在residual block中,通过设置卷积参数,可以控制residual block的输入和输出的特性一致,从而进行相加,避免了解决深度网络的梯度消失和梯度下降问题。ResNet的结构大部分是1层Conv+4blocks+1层fc,分为5个stage,请查看我们Word文档中的表格。一开始,ResNet对输入进行预处理,即表中第一行,(256*256*3)代表输入的宽高和通道数。在这个阶段,

通过大小为7*7、数量为64的卷积核,经过Batch Normalization后,使用ReLu激活函数进行激活。

这个阶段的第二层是最大池化层。最大池化层通过大小为3*3、步长为2的核进行卷积运算,输出图像大小为(64*64*64),其中64为第一个卷积层的卷积核数。一般来说,在这个阶段,shape为(256*256*3)的输入经过卷积层、BN层、ReLU激活函数、MaxPooling层得到(64*64*64)的输出。

经过上面的描述,就更容易理解表中的后续阶段了。瓶颈由 4 个可变参数组成,通道数以及长度和宽度。将输入的形状X定义为(C,W,W)通过BN和ReLU作为卷积函数F(x),将两者加到F(x)+x之后,通过一个ReLU函数得到输出,并且输出的形状仍然是(C,W,W)。通过这个操作,匹配输入和输出的维度差,然后求和。

要点3:  表4与表3相同,未给出不同分类模型的分类结果。

回复3:感谢您的指正。我们已经修改了这张表,请查看此处或查看我们最新的手稿修订版。(第 15 页)

表 4. 不同网络算法的故障诊断测试结果。

方法

参数个数/个

训练时间/s

准确性/%

支持向量机

458356

253

76.34

英国石油公司

1267423

532

72.41

1DCNN+信号

15467261

6458

78.56

GADF+VGG16

136145681

34436

85.72

GADF+GhostNet

4724368

3524

91.54

GADF+Resnet50

26541072

15241

93.22

GADF-SEResnet

25328635

13862

98.37

Reviewer 3 Report

1. The title of the article cannot highlight innovative points, it is recommended to choose a new title.

2. Since your proposed algorithm aims to solve the problem of timeliness, in the introduction, you need to introduce some time-consuming articles. And explain the reason why these models take a long time. Afterwards, you need to introduce the proposed algorithm again. Express what improvements you have made and why your proposed algorithm consumes less time.

3. I don't think the second part of the article is representative. A summary should be provided on the issues to be addressed. It is very abrupt to only write about feature extraction and feature extraction in fault diagnosis. There is no progressive relationship with the experiment you conducted.

4. Reference [13] cited incorrectly.

5. In Figure 4, it is recommended to unify the font. The annotations in Table 2 and Figure 6 also have special fonts. Please review the entire text for consistency.

6. Section 3.5 suggests explaining the setting of relevant parameters.

7. It is recommended to replace 8:2 in 8.2.1 with 4:1.

8. The content of Table 4 is filled in incorrectly, which is the same as Table 3.

9. In the final experimental section, you first demonstrated the superiority of GADF. Secondly, if you want to demonstrate the superiority of the improved model, you cannot directly use one-dimensional vibration input to SVM and BP networks.

10. In addition, you compared the timeliness and only limited the image forwarding method. The runtime of different models should also be added in Table 4.

Author Response

Response to Reviewer 3 Comments

Point 1: The title of the article cannot highlight innovative points, it is recommended to choose a new title.

Response 1: Thank you for your kind suggestion. We have change the title as A Deep Learning Method for Rolling Bearing Fault Diagnosis Based on Attention Mechanism and Graham Angle Field.

Point 2:  Since your proposed algorithm aims to solve the problem of timeliness, in the introduction, you need to introduce some time-consuming articles. And explain the reason why these models take a long time. Afterwards, you need to introduce the proposed algorithm again. Express what improvements you have made and why your proposed algorithm consumes less time.

Response 2: Thank you for your correction. We have revise the related work part(Page 4-5) and try to make it more methodical. Actually when we first have the idea of this manuscript we want to find a way to transform the signal into image so that we can find some clue through the diagnosis process. We consider that only one kind of signal maybe can identify one part of the mistake but for the equipment in factory there are various kinds of input signal. We want to find the assosiation between the signal and the data so we try to use the GADF and ResNet for fault diagnosis and during that we found this maybe a good idea to identify the fault of bearing which is widely used and significently for the equipment. In the future we will find more deep assiastion between the signal and image. So we can combine the heterogeneous multi-source data and discover the relationship.

The latest part of Section 2 is follow or please check out on our latest revision:

2.2. Data Feature Extraction in Fault Diagnosis

From the point of view of industry and academia, early detection of equipment failures generally monitors equipment operation data and uses feature extraction to extract information related to equipment health status from complex equipment oper-ation data[34,35]. Especially in recent years, with the widespread use of sensors, the sampling frequency of various signals such as vibration, temperature and pressure has increased, in order to diagnose the healthy operation status of equipment and finally achieve the purpose of predictive maintenance[36], it is required to data analysis, then the inevitable problem of the disaster of data dimensionality will arise in the process. Faced with this kind of dimensionality disaster problem, traditional machine learning methods do not perform well. The common idea is to reduce the dimensionality of da-ta before data input or into the model, using methods such as artificial neural network (ANN)[37], support vector machine(SVM)[38], etc. , so as to achieve better results. However, these traditional intelligent fault diagnosis methods often have poor gener-alization ability of sensitive features and excessive reliance on prior knowledge of ex-perts when dealing with massive data. That is to say, for each specific fault diagnosis task, the feature extractor must be redesigned. , therefore, it is an urgent need to re-search a feature selection method that can eliminate manual extraction.

In view of the above, it can be found that CNN has made achievements in the field of image classification, and deep learning as a feature selection method has also been widely used in equipment fault diagnosis.

For example, Shao et al.[39] designed a deep belief network (DBN) for rolling bearing fault diagnosis. They used the idea of particle swarm optimization to greatly improve the accuracy of rolling bearing fault diagnosis, and obtained a calcula-tion result of 13s. In addition, Chen et al[40] used sparse encoders for fault diagnosis by fusing the time domain and frequency domain features of the vibration signal. This method has a high recognition accuracy, and the accuracy rate reaches about 98%. Based on the neural network has a stronger advantage in image classification, it has become the focus of discussion by many scholars to present the sensor signal through a certain encoding method and use the neural network for feature processing. Among them, Lu et al[41] The signal is converted into an image using a bispectrality method to classify the image using a neural network. With 98% accuracy and robustness, the method finally achieves fault classification. Yin et al.[42] used a data generation strat-egy based on generative adversarial networks and convolutional neural networks to perform fault analysis on vibration signals under different noise environments, and achieved good results. However, the speed decreases significantly with the increase of sample size.

Although CNN has demonstrated extremely high advantages in image feature extraction, there is still room for further improvement in the problem of feature extrac-tion from two-dimensional images converted from bearing vibration signals proposed in this paper.

First, the Graham angle field diagram obtained by passing the original vibration signal through the Graham angle field is used as the input vector feature of the neural network. But this kind of simple signal conversion as a one-dimensional input of fault diagnosis still can't get high-dimensional information. In order to better understand and mine the high-dimensional information of faults and improve the accuracy and speed of neural network recognition, this paper introduces the attention mechanism in ResNet. In the process of feature extraction, we refer to the SeBlock module. Through this feature re- According to the influence of feature channels on model performance, different weight factor values are assigned to different feature channels.

Through this method, the discriminative feature channels in the image are high-lighted, and the inconspicuous features in the image are suppressed, thereby enhanc-ing the pertinence of the model to specific channels, and finally improving the fine feature extraction and local area localization capabilities of the model.

Point 3:  I don't think the second part of the article is representative. A summary should be provided on the issues to be addressed. It is very abrupt to only write about feature extraction and feature extraction in fault diagnosis. There is no progressive relationship with the experiment you conducted.

Response 3: Thank you for your suggestion. We have change the Section 2 and try to make it clear for readers to understand the purpose and the progressive relationship between the related work and the experiment we design. Please check out our latest vision(Page5) or read as follow:

Although CNN has demonstrated extremely high advantages in image feature extraction, there is still room for further improvement in the problem of feature extraction from two-dimensional images converted from bearing vibration signals proposed in this paper.

First, the Graham angle field diagram obtained by passing the original vibration signal through the Graham angle field is used as the input vector feature of the neural network. But this kind of simple signal conversion as a one-dimensional input of fault diagnosis still can't get high-dimensional information. In order to better understand and mine the high-dimensional information of faults and improve the accuracy and speed of neural network recognition, this paper introduces the attention mechanism in ResNet. In the process of feature extraction, we refer to the SeBlock module. Through this feature re- According to the influence of feature channels on model performance, different weight factor values are assigned to different feature channels.

Through this method, the discriminative feature channels in the image are highlighted, and the inconspicuous features in the image are suppressed, thereby enhancing the pertinence of the model to specific channels, and finally improving the fine feature extraction and local area localization capabilities of the model .

Point 4: Reference [13] cited incorrectly.

Response 4: Thank you for your comment. We have recited this one. It is our fault to make this happen.

Point 5:  In Figure 4, it is recommended to unify the font. The annotations in Table 2 and Figure 6 also have special fonts. Please review the entire text for consistency.

Response 5: Thank you for your recommend. We have make this correct and check the format of all the Tables and Figures. It is our fault to make this happen. Please check out our latest vision of manuscript.

Point 6:  Section 3.5 suggests explaining the setting of relevant parameters.

Response 6: Thank you for your suggestion. We have added this part in Section 3(Page 7-9), please check out our latest revision or read as follow:

The residual block structure can be divided into two types: Basic Block and Bottleneck Block according to different application environments, as shown in Figure 3. By introducing a shortcut branch, the low-level information is shared to the high-level through layer-skip connections, which solves the problem that the gradient cannot be optimized during the backpropagation of the deep network, and breaks through the model degradation and gradient anomalies that limit the number of model layers.

For different residual block structures, Bottleneck Block has fewer parameters than Basic Block by changing the number of channels, and is more economical in terms of calculation, and is often used in networks with deeper layers.

According to the difference in residual block structure, ResNet can be divided into the following common models: ResNet18, ResNet 34, ResNet50, ResNet101, ResNet152. Considering the device configuration and the difficulty of image training, this article uses ResNet50 as the basic model, and the parameters are shown in Table 1 and Figure 2. Please check out the table and figre in the Word file.

ResNet structure diagram can refer to the following figure. A 7*7 convolution kernel is responsible for feature extraction, and the convolution step is 2 so that the length and width of the image are first reduced to 1/2 of the previous one. Then pass a MaxPooling layer to further reduce the resolution of the image. Please check out the figures in the Word file.

By using the residual block, the number of input feature map channels is doubled.

Each stage will consist of a down sampling block and two residual blocks. The down sampling is set to the initial convolution step size, which is 2, in this way, the length and width can be reduced. In the residual block, by setting the convolution parameters, the characteristics of the input and output of the residual block can be controlled to be consistent, so that they can be added, which avoids solving the gradient disappearance and gradient descent problems of deep networks. Most of the structure of ResNet is 1 layer Conv+4 blocks+1 layer fc, which is divided into 5 stages, as shown in the Table 1 line2 to line8. Please check out the table in the Word file. At the very beginning, ResNet preprocesses the input, that is, the first line in the table, (256*256*3) represents the input width, height, and number of channels. At this stage, it includes the following three successive operations.

Through the convolution kernel with a size of 7*7 and 64, after Batch Normalization, it is activated using the ReLu activation function.

The second layer at this stage is the maximum pooling layer. The maximum pooling layer performs convolution operations through a kernel with a size of 3*3 and a step size of 2. The output image size is (64*64*64) where 64 is the number of convolution kernels in the first convolutional layer. In general, at this stage, the input with a shape of (256*256*3) passes through the convolutional layer, BN layer, ReLU activation function, and MaxPooling layer to obtain an output of (64*64*64).

After going through the above description, it is easier to understand the subsequent stages in the table. The bottleneck consists of 4 variable parameters, the number of channels and the length and width. The input shape X is defined as (C, W, W) through BN and ReLU as the convolution function F (x), and the two are added to F (x) + x, the output is obtained through a ReLU function, and the shape of the output is still (C, W, W). Through this operation, the dimension difference between the input and the output is matched, and then summed.

Point 7:  It is recommended to replace 8:2 in 8.2.1 with 4:1.

Response 7: Thank you for your comment. For the traditional machine learning stage (the data set is on the order of 10,000), the general distribution ratio is 7:3 or 8:2 for the training set and test set.

However, this division method is indeed arbitrary. In order to further test the suitability of the divided data set, the obtained recall rate and accuracy are shown in the following table:

Training

Precision

Recall

0.8

0.3895

0.1652

0.6

0.3744

0.1370

0.4

0.3686

0.1005

0.2

0.2563

0.0834

According to the above results, we choose to use a training test ratio of 8:2 to get the best results.

Point 8:  The content of Table 4 is filled in incorrectly, which is the same as Table

Response 8: Thank you for your correction. We have revised this part and it is our fault to make this happen. Please check out our latest vision of manuscript(Page 15) or read as follow:

Table 4. Fault Diagnosis Test Results of Different Network Algorithms.

Method

Number of parameters/pieces

Training time/ms

Accuracy/%

1DCNN+Signal

15467261

6458

78.56

GADF+VGG16

136145681

34436

85.72

GADF+GhostNet

4724368

3524

91.54

GADF+Resnet50

26541072

15241

93.22

GADF- SEResnet

25328635

13862

98.37

Point 9: In the final experimental section, you first demonstrated the superiority of GADF. Secondly, if you want to demonstrate the superiority of the improved model, you cannot directly use one-dimensional vibration input to SVM and BP networks.

Response 9: Thank you for your comment. It is indeed an unwise choice to use the vibration input to the SVM and BP network for a comprehensive comparison(Page 15). In order to be more rigorous, we deleted these two sets of comparison data, and only kept the comparison related to CNN. Thank you for your opinion.

Point 10:  In addition, you compared the timeliness and only limited the image forwarding method. The runtime of different models should also be added in Table 4.

Response 10: Thank you for your correction. We have revised this part and it is our fault to make this happen. Please check out our latest vision of manuscript(Page 15).

Round 2

Reviewer 3 Report

1. It is better to delete 2.1 and directly describe 2.2 Feature extraction in fault diagnosis and give some examples.

 2. Figure 6 still needs to be double-checked. The beginning and the end should be indicated in the ellipse circles. Please check if 'Converge or not' is a pen error. Should 'Test set' correspond to 'Testing set'.

 3. Section III Materials and Methods is misplaced in format.

 4. The 15th, 27th, 29th, 32nd, 33rd, 34th, 38th, 45th, and 49th reference years are not bolded.

Author Response

Response to Reviewer 3 Comments

Point 1: It is better to delete 2.1 and directly describe 2.2 Feature extraction in fault diagnosis and give some examples.

Response 1: Thank you for your correction. We have read the Section 2 carefully and there do have some inappropriate between the main body and the literature review. We have added some more examples and try our best to fix this part. Please check out the latest revision or read as follow:

  1. Literature review

2.1. Data Feature Extraction in Fault Diagnosis

From the point of view of industry and academia, early detection of equipment failures generally monitors equipment operation data and uses feature extraction to extract information related to equipment health status from complex equipment operation data[13,14]. Especially in recent years, with the widespread use of sensors, the sampling frequency of various signals such as vibration, temperature, and pressure has increased, in order to diagnose the healthy operation status of equipment and finally achieve the purpose of predictive maintenance[15], it is required to data analysis, then the inevitable problem of the disaster of data dimensionality will arise in the process. Faced with this kind of dimensionality disaster problem, traditional machine learning methods do not perform well. The common idea is to reduce the dimensionality of data before data input or into the model, using methods such as artificial neural network (ANN)[16], support vector machine(SVM)[17], etc. , so as to achieve better results. However, these traditional intelligent fault diagnosis methods often have poor gener-alization ability of sensitive features and excessive reliance on prior knowledge of experts when dealing with massive data. That is to say, for each specific fault diagnosis task, the feature extractor must be redesigned. , therefore, it is an urgent need to research a feature selection method that can eliminate manual extraction.

Starting from the traditional autoencoder (AE), many scholars have tried various methods in the field of fault diagnosis for fault feature recognition. For example, Jia et al.[18] used a method based on DNN (deep neural network) to find a nonlinear function that can approximate complex nonlinear functions from the original data. , so as to find a method for automatic and accurate fault diagnosis in this way. In addition, Xia et al.[19] are based on DAE (Denoising Autoencoder) and CAE (Shrinkage Autoencoder). A high-quality fault diagnosis method that can not only locally preserve projections but also fuse deep features is created, and the proposed method is more effective and robust than traditional signal analysis methods. Similarly, Liu et al.[20] used stacked sparse AE, they combined short-time Fourier transform and stacked sparse autoencoder to analyze the signal, through ablation After the experiment, it is found that this method has better performance and effectiveness. Recently, in order to better use deep learning for fault analysis of signals, Jia et al.[21] based on the shortcomings of traditional autoencoders, by constructing a normalized locally connected network of the sparse autoencoder, using the special rules learned from the input signal in the local layer, finally obtains the recognition health result by obtaining the translation in-variant features, and effectively recognizes the health state of the machine.

After CNN was proposed, due to its advantages in extracting sensitive features from high-dimensional data, some scholars were inspired to convert vibration signals into two-dimensional images and use CNN for feature extraction, so as to finally obtain the distribution status of faults. For example, Guo et al.[22] proposed an adaptive CNN network structure, which can effectively identify the features in the sample in the data set and automatically extract deep features, the experimental verification found that it has better performance. At the same time, Zhang et al.[23] proposed a convolutional neural network with data enhancement effects, using the width of the first convolutional layer The kernel implements the adaptability of the model with high accuracy.

In view of the above, it can be found that CNN deep learning has made achievements in the field of image classification, and deep learning as a feature selection method has also been widely used in equipment fault diagnosis.

For example, Shao et al.[24] designed a deep belief network (DBN) for rolling bearing fault diagnosis. They used the idea of particle swarm optimization to greatly improve the accuracy of rolling bearing fault diagnosis and obtained a calculation result of 13s. In addition, Chen et al[25] used sparse encoders for fault diagnosis by fusing the time domain and frequency domain features of the vibration signal. This method has a high recognition accuracy, and the accuracy rate reaches about 98%. Based on the neural network having a stronger advantage in image classification, it has become the focus of discussion by many scholars to present the sensor signal through a certain encoding method and use the neural network for feature processing. Among them, Lu et al[26] The signal is converted into an image using a bispectrality method to classify the image using a neural network. With 98% accuracy and robustness, the method finally achieves fault classification. Yin et al.[27] used a data generation strategy based on generative adversarial networks and convolutional neural networks to perform fault analysis on vibration signals under different noise environments, and achieved good results. However, the speed decreases significantly with the increase in sample size.

Although CNN has demonstrated extremely high advantages in image feature extraction, there is still room for further improvement in the problem of feature extraction from two-dimensional images converted from bearing vibration signals proposed in this paper.

First, the Graham angle field diagram obtained by passing the original vibration signal through the Graham angle field is used as the input vector feature of the neural network. But this kind of simple signal conversion as a one-dimensional input of fault diagnosis still can't get high-dimensional information. In order to better understand and mine the high-dimensional information of faults and improve the accuracy and speed of neural network recognition, this paper introduces the attention mechanism in ResNet. In the process of feature extraction, we refer to the SeBlock module. Through this feature re-According to the influence of feature channels on model performance, different weight factor values are assigned to different feature channels.

Through this method, the discriminative feature channels in the image are highlighted, and the inconspicuous features in the image are suppressed, thereby enhancing the pertinence of the model to specific channels, and finally improving the fine feature extraction and local area localization capabilities of the model.

Point 2: Figure 6 still needs to be double-checked. The beginning and the end should be indicated in the ellipse circles. Please check if 'Converge or not' is a pen error. Should 'Test set' correspond to 'Testing set'.

Response 2: Thank you for your comments. We have changed the Figure 6 please check out our latest vision or read as follows:

Point 3: Section III Materials and Methods is misplaced in format.

Response 3: Thank you for your correction. We have carefully revised the format of Section III(Page 4). Please check out our latest vision of the manuscript.

Point 4: The 15th, 27th, 29th, 32nd, 33rd, 34th, 38th, 45th, and 49th reference years are not bolded

Response 4: Thank you for your comment. Due to the revision of Section 2, there are few references have been replaced but we have changed the format(Page 16) so there may be a little bit of confusion about the serial number of reference right now between your suggestion. Please check out our latest revision of the manuscript.
